# EMOE: Expansive Matching of Experts for Robust Uncertainty Based Rejection

## Abstract

Expansive Matching of Experts (EMOE) is a novel method that utilizes support-expanding, extrapolatory pseudo-labeling to improve prediction and uncertainty based rejection on out-of-distribution (OOD) points. We propose an expansive data augmentation technique that generates OOD instances in a latent space, and an empirical trial based approach to filter out augmented expansive points for pseudo-labeling. EMOE utilizes a diverse set of multiple base experts as pseudo-labelers on the augmented data to improve OOD performance through a shared MLP with multiple heads (one per expert). We demonstrate that EMOE achieves superior performance compared to state-of-the-art methods on both image and tabular data.

## 1 Introduction

It is well-known that the generalization capabilities of models can be severely limited when tested on out-of-distribution (OOD) data that deviates from the distribution seen at training time (Torralba and Efros, 2011; Liu et al., 2021; Freiesleben and Grote, 2023). This in turn affects many real-world applications where models may be evaluated on distribution-shifted data during deployment. For instance, these issues commonly arise in medical applications where patient distributions at inference time may deviate from the training data (Lee et al., 2023). A potential strategy for the safe deployment of models in real-world applications is to employ novelty-based rejection (Dubuisson and Masson, 1993; Hendrickx et al., 2024), where predictions are rejected whenever the model is evaluated on an instance that deviates from the data distribution seen during training. While such an approach is appropriate in certain scenarios (for example, whenever it is expected that a human will be in the loop and thus can easily intervene upon rejection), this prevalent strategy is overly-conservative as it foregoes any potential extrapolation[1] by design. That is, novelty-rejection forbids any form of extrapolation (predictions outside of the training data sup-

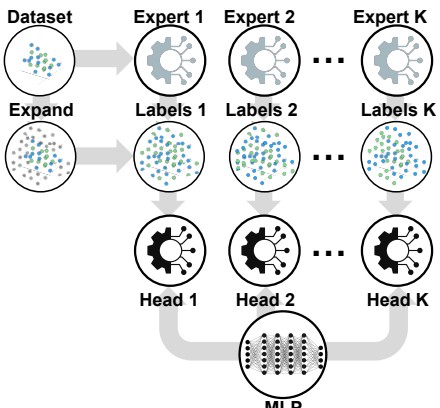

Figure 1: *Mockup*. Our approach trains a set of diverse base experts on training data. After, we consider a novel augmentation to expand the distributional support in a latent space. Then, using a shared neural network (MLP) with multiple heads (one per expert) we train the network to match the expert labels on the expanded data.

port), even when the model may be capable. Instead, in this work we attempt to *better assess* (and improve) the limits of models' ability to extrapolate based on dataset augmentation and self-training on a set of experts (see Fig. 1).

**Broader Impacts** As a motivating application, consider the use of ML models for scientific *discovery*. In such discovery applications, ML models should, by definition, characterize data that is novel and distinct from what has been previously characterized. These, however, are exactly the sort of inferences that are disallowed by novelty-rejection methods, which essentially forbid any

---

[1]We use the term extrapolation to loosely encompass prediction outside of the training data distribution support (without consideration of the data's convex-hull).

potential characterization of instances that expand the support of characterized data (*regardless of a model's capability to do so*). Take, for instance, drug discovery (a driving application for this paper) where one hopes to predict desirable properties of molecules that are quite distinct from molecules that have been previously characterized. (I.e., "scaffold hoping" (Hu et al., 2017), leveraging existing data to discover a desirable molecule with significantly different chemical structure.) Novelty-based rejection would reject any prediction on molecules that do not fall firmly in the support of what has been previously characterized and used for model training. Therefore, these typical rejection techniques only allow predictions on molecules that are similar to those already characterized in the training set (akin to the notion of 'interpolation') preventing their utility in discovering structurally novel molecules. Instead, we wish gain a better understanding of *what* OOD ('extrapolatory') instances may be well predicted by our model. In drug discovery, the use of models with poor confidence filtration will result in *false positives* that waste resources on unsuccessful experiments where screened molecules do not present desirable properties (see Fig. 2); hence, it is key that high-confidence predictions on OOD instances correspond with accuracy (i.e., that confidence filtration on OOD instances leads to higher-quality predictions).

**Contributions** Unfortunately, in-distribution confidence based measurements fail to properly characterize model capabilities on OOD data (Arjovsky et al., 2019; Creager et al., 2020). In this work we show that we can improve confidence-based filtration of predictions (as measured by area under precision-recall and receiver operating characteristic curves, AURPC/AUROC) on OOD instances with a novel training scheme of a multi-headed network based on matching with augmented (expanded) data (see Fig. 1, Alg. 1). In particular, our contributions are as follows: 1) we propose a novel expansive data augmentation technique that generates OOD instances in a latent space; 2) we propose a novel empirical trial based approach to filter out augmented expansive points for psuedo-labeling; 3) we develop a straight-forward but effective strategy that yields a strong, diverse set of base-experts for self-training; 4) we develop our novel EMOE approach for training a multi-headed network; 5) we show state-of-the-art (SOTA) performance in rejecting predictions via estimated confidences via AUPRC-based metrics (see Fig. 3) in a single-source generalization setting (Qiao et al., 2020).

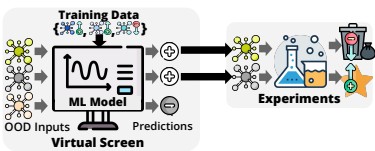

(a) Poor Extrapolative Rejection

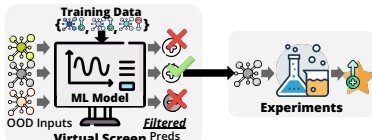

(b) Filtration on Extrapolative Data.

Figure 2: *Virtual screening.* (a) Often, ML models yield unreliable predictions that will waste resources on unsuccessful experiments (Kimber et al., 2021). (b) We seek reliable extrapolatory predictions (✓) for better use of experimental resources.

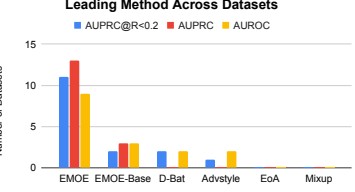

Figure 3: Tally of datasets where respective methods lead in metrics: AUPRC for recall less than .2 (AUPRC@R< .2), AUPRC, and AUROC. (See further details in §4.)

---

**Algorithm 1** Expansive Matching of Experts (EMOE) Approach

---

1: Learn a latent space.
2: Train a set of base experts.
3: Expand training set with data that falls beyond the support in the latent space.
4: Train a network with multiple heads (one per base expert) matching the experts on expanded and original training dataset.
5: Infer with a combination of the base experts and network heads.

---

## 2 RELATED WORK

**Domain Generalization** Domain generalization (DG) aims to learn a model that is able to generalize to multiple domains. A typical approach is to learn a domain invariant representation across

multiple source domains. Domain invariant representation learning can be done by minimizing variations in feature distributions (Li et al., 2018; Ding et al., 2022) and imposing a regularizer to balance between predictive power and invariance (Arjovsky et al., 2019; Koyama and Yamaguchi, 2020). Another line of research incorporates data augmentation to improve generalizability. Basic transformations like rotation and translation, varying in magnitude, are commonly used on images to diversify the training data (Cubuk et al., 2019; Berthelot et al., 2020). More sophisticated augmentation techniques have recently surfaced: (Zhang et al., 2018) introduced mixup, which linearly combines two training samples; (Yun et al., 2019) proposed CutMix, blending two images by replacing a cutout patch with a patch from another image; (Zhong et al., 2022) adversarially augment images to prevent over fitting to source domains. We focus on augmentations that are general and applicable across modalities.

**Self-Training**   Self-training aims to utilize an earlier model as a pseudo-labeler to populate the training data with more labelled instances by labelling unlabelled data at each iteration. Then, the new labelled set is combined with the previous training set to train a new model. The concept of pseudo-labeling was initially proposed by (Lee, 2013), suggesting a straightforward approach of retaining instances where the teacher model has high prediction probabilities. Following (Lee, 2013; Zou et al., 2018) proposed to use a proportion of the most confident unlabelled data points instead of a fixed threshold. Researchers then combined curriculum learning with pseudo labeling, where thresholds for acquiring unlabeled data for each class is dynamically adjusted at different time steps, allowing the most informative unlabeled data to be incorporated (Cascante-Bonilla et al., 2020; Zhang et al., 2021). Another line of research improves robustness of pseudo-labeling by encouraging diversity in the pseudo-labeler. In their work, Ghosh et al. (2021) employed an ensemble of models as teacher to provide pseudo-labels for the student model. On the other hand, (Xie et al., 2019) injected noise into the pseudo-labeler model by incorporating Dropout (Srivastava et al., 2014) and data augmentation techniques to provide more robust pseudo-labels. In this work we show how to utilize multiple pseudo-labelers on extrapolatory augmented data to improve OOD performance.

**Selective Classification**   Reject option methods (also known as selective classification) aim to identify instances where the model should not predict. Many selective classification approaches rely on a post-training processing strategy. Following this strategy, once the model has finished training, a rejection metric is computed. Then, predictions are rejected or accepted based on a predefined threshold. A simple choice for a rejection metric is to utilize the conditional output probability from ML models (Stefano et al., 2000; Fumera et al., 2000). Building upon these works, (Devries and Taylor, 2018) proposed to train a confidence branch alongside of the prediction branch by incentivizing a neural network to produce a confidence measure during training; Geifman and El-Yaniv (2017) proposed a method for constructing a probability-calibrated selective classifier with guaranteed control over the true risk. Recently, methods adopting end-to-end training approaches have been proposed (Thulasidasan et al., 2019) (Ziyin et al., 2019) (Geifman and El-Yaniv, 2019). In these works, an extra class is added when predictions are made. If the extra class has the highest class probability for a sample, the sample is rejected. Most reject-option approaches are geared towards in-distribution rejection and utilize novelty-rejection when encountering any OOD points (Torralba and Efros, 2011; Liu et al., 2021; Freiesleben and Grote, 2023); instead, we propose to learn better conditional output probabilities on OOD data for more effective, capability-aware rejection.

**Ensemble Modeling**   Ensemble techniques aim to utilize a diverse set of models jointly for better performance. Early methodologies for ensembles aggregate (bag) predictions from all models (Dieterich, 2007)(Kussul et al., 2010) or a subset of the models in the ensemble (Jordan and Jacobs, 1993), (Eigen et al., 2013). In the OOD setting, prior works addressed this problem by enforcing prediction diversity on OOD data (Pagliardini et al., 2023), ensembling moving average models (Arpit et al., 2022a), and training an ensemble of domain specific classifiers (Yao et al., 2023). EMOE adopts a multi-headed architecture that produces an ensemble to improve predictions on OOD data.

## 3   METHOD

**Training Data**   Throughout, we assume the '*single-source*' generalization setting (Qiao et al., 2020), where we observe a single in-distribution (ID) training dataset $\mathcal{D} = \{(x_i, y_i)\}_{i=1}^{N}$, and instances are drawn *iid* $(x_i, y_i) \sim \mathcal{P}_{\text{in}}$ *without* any accompanying environmental/domain/source information *nor any labeled/unlabeled OOD instances*. For simplicity, we write to the binary classification case,

$y_i \in \{0, 1\}$, but our methodology is easily extendable to other supervised tasks. We design our method to work in general, non-modality specific[2] (e.g., image, text, audio) settings such as the real-valued case $x_i \in \mathbb{R}^d$.

**Base Collection of Experts** EMOE leverages a set of diverse initial experts $\{g_k\}_{k=1}^K$ to guide the training of a secondary model. There are many mixture of experts (Jordan and Jacobs, 1993), (Eigen et al., 2013) (Du et al., 2021) and ensembling (Arpit et al., 2022a) (Dietterich, 2007) (Pagliardini et al., 2023) (Yao et al., 2023) methods available; we observe good empirical performance (see Sec. 4) using a collection of strong base-learners trained on uniform sub-selections of instances *and* features (akin to the construction of a random forest).

### 3.1 EXPANSIVE AUGMENTATION OF TRAINING DATA

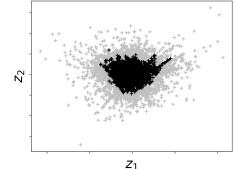

We begin with the simple intuition that if we want to improve extrapolatory performance of models, then we should consider training signals on instances that lie outside of the original data support. While there has been much recent attention in strong augmentations to improve OOD performance in modality-specific settings (Zhong et al., 2022) (Xie et al., 2019), performing such augmentations on general data remains a challenge. To reason about the support of the training data, and how to *expand* past it, we propose to leverage a latent factor space, $\varphi : \mathbb{R}^d \mapsto \mathbb{R}^s$. While learning semantically meaningful latent factor spaces remains an active area of research, we observed strong performance utilizing autoencoding techniques (see Sec. 4), which carry a corresponding decoder $\gamma : \mathbb{R}^s \mapsto \mathbb{R}^d$. Without loss of generality, we consider centered latent spaces such that $\mathbb{E}[\varphi(X)] = 0$.

Figure 4: Expansion in latent space: points (black) are augmented (gray) and expand the distributional support.

We propose a novel, yet straightforward strategy to expand data outside of training distributional support: perturb instances to lie further away from the origin in latent space. In particular, if we have latent vector $z = \varphi(x)$, we propose to consider perturbations of the form $z' = (1 + |\epsilon|)z$ where $\epsilon \sim \mathcal{N}(0, 1)$, and one can utilize the decoder $x' = \gamma(z')$. That is, we define our expansion operation on a set of points as:

$$\mathbf{Ex}(\{x_i\}_{i=1}^N) \equiv \{\gamma\left((1 + |\epsilon_i|)\varphi(x_i)\right) \mid \epsilon_i \sim \mathcal{N}(0, 1)\}_{i=1}^N. \tag{1}$$

**Ex** will be a *stochastic* mapping. Clearly, the perturbed latent codes will tend away from areas of support (see Fig. 4). However, unlike with small jitter-based perturbations, where one can retain an original instance label, it is less clear how to derive an accompanying training signal for expansive augmentations. Here, we propose to leverage a pseudo-labeling scheme where we derive $K$ labels with the base experts $(f_1(x'), \ldots, f_K(x'))$[3]. We expound on utilizing the base expert labels below.

### 3.2 TRUSTWORTHY EXPANSIVE SIGNALS WITH EXTRAPOLATORY DIRECTIONAL MINING

Recent works have noted that confidence based filtration of pseudo-labels improves self-training techniques (Lee, 2013; Sohn et al., 2020). Here we present a novel, complementary approach to filter out pseudo-labels on expansive augmented data (as above). We wish to filter out expansive augmentations where predicted labels may not be accurate or helpful. Of course, predicting the quality of extrapolatory performance is in itself a difficult task, since we do no have access to any OOD data in our setting. Our approach is based on empirical *non-iid* held-out trials to judge the difficulty of labeling extrapolatory instances in some direction (in latent space), and keep track of directions that are easiest to extrapolate to for later augmentation. That is, for a given direction on the hyper-sphere $v \in \mathbb{S}^{s-1}$, we withhold the training instances that have the highest projections w.r.t. $v$ (are in the highest $q$-th quantile), $\mathcal{T}_v^{\text{held}} \equiv \{(x, y) \in \mathcal{D} \mid \varphi(x)^T v \ge t_{v,q}\}$, where $t_{v,q}$ is the threshold of the $q$-th quantile for projections of training latent vectors onto $v$. The remaining points in the training set are then used to train a model for our trial on direction $v$: $\mathcal{T}_v^{\text{train}} \equiv \mathcal{D} \setminus \mathcal{T}_v^{\text{held}}$. We train a simple parametric model (e.g., a linear logistic-regression model) on $\mathcal{T}_v^{\text{train}}$, $f_v$ and evaluate a performance metric, $\rho$ (e.g., accuracy, F1-score, etc.), on the held-out trial data $\rho(\mathcal{T}_v^{\text{held}}, f_v)$. We take well-performing trials (based on $\rho(\mathcal{T}_v^{\text{held}}, f_v)$) as evidence that extrapolation to extreme values

---

[2]In particular, we avoid any modality or domain-specific augmentation of data.

[3]One may also train base experts directly on the latent space, $(f_1(z'), \ldots, f_K(z'))$, and avoid the decoder.

in $v$ is possible with the data, and hence accrue instances in $\mathcal{T}_v^{\text{held}}$ into a large collection $\mathcal{E}$ for pseudo-labeling with the base experts (see Alg. 2 for details).

---

**Algorithm 2** Extrapolatory Directional Mining

1: **procedure** GET_DIRECTIONAL_EXPANSION_POINTS($\varphi, M, T, q, \mathcal{D}$)          ▷ get quality points to expand, $\mathcal{E}$, based on top $M$ (of $T$) performing (based on metric $\rho$) directional empirical held-out trials withholding the $q$-th percentile of projections in latent space $\varphi$.
2:     `heap.init()`
3:     **for** $j \in \{1, \ldots, T\}$ **do**
4:         $v \leftarrow \text{Unif}(\{\frac{x}{\|x\|} \mid (x, y) \in \mathcal{D}\})$                    ▷ random direction
5:         $t_{v,q} \leftarrow \texttt{quantile}(\{\varphi(x)^T v \mid (x, y) \in \mathcal{D}\}, q)$
6:         $\mathcal{T}_v^{\text{held}} \leftarrow \{(x, y) \in \mathcal{D} \mid \varphi(x)^T v \geq t_{v,q}\}$        ▷ withhold extreme points on direction
7:         $\mathcal{T}_v^{\text{train}} \leftarrow \mathcal{D} \setminus \mathcal{T}_v^{\text{held}}$.                        ▷ train on rest
8:         $f_v \leftarrow \texttt{model.fit}(\mathcal{T}_v^{\text{train}})$
9:         $\texttt{heap.push}(\rho(\mathcal{T}_v^{\text{held}}, f_v), \mathcal{T}_v^{\text{held}})$          ▷ order based on performance
10:     **end for**
11:     $\mathcal{E} = [\,]$
12:     **for** $j \in \{1, \ldots, M\}$ **do**                          ▷ get points in top $M$ trials
13:         $\mathcal{T} \leftarrow \texttt{heap.pop()}$
14:         $\mathcal{E}\texttt{.append}(\{x \mid (x, y) \in \mathcal{T}\})$
15:     **end for**
16:     **return** $\mathcal{E}$
17: **end procedure**

---

### 3.3 MATCHING NETWORK WITH EXPERTS ON EXPANSIVE DATA

We propose learning a multi-headed network based on self-training with base experts' predictions on the (filtered) expanded set of training data, $\mathcal{E}$. That is, we propose learning a network composed of a shared multilayer perceptron (MLP), $\phi : \mathbb{R}^d \mapsto \mathbb{R}^m$, and $K$ expert matching heads, $h_1, \ldots, h_K$; e.g., mapping to logit space for binary classification, $h_j : \mathbb{R}^m \mapsto \mathbb{R}$. Our loss incorporates a per head loss that matches experts on a set $\mathcal{S}$:

$$\mathcal{L}_{\text{match}}(\phi, \{h_j\}_{j=1}^K, \{g_j\}_{j=1}^K; \mathcal{S}) \equiv \frac{1}{|\mathcal{S}|K} \sum_{x \in S} \sum_{j=1}^K \ell(h_j(\phi(x)), g_j(x)), \tag{2}$$

where $\ell(\hat{y}, y)$ is a supervised loss (e.g., the cross-entropy loss). Moreover, we will utilize a mean-matching L1 loss

$$\mathcal{L}_{\text{mean}}(\phi, \{h_j\}_{j=1}^K, \{g_j\}_{j=1}^K; \mathcal{S}) \equiv \frac{1}{|\mathcal{S}|} \sum_{x \in S} \left\| \frac{1}{K} \sum_{j=1}^K \sigma(h_j(\phi(x))) - \frac{1}{K} \sum_{j=1}^K g_j(x) \right\|_1, \tag{3}$$

where $\sigma(\cdot)$ is the sigmoid. Our full expansive matching of experts loss is then:

$$\mathcal{L}_{\text{EMOE}}(\phi, \{h_j\}_{j=1}^K, \{g_j\}_{j=1}^K; \mathcal{D}, \mathbf{Ex}(\mathcal{E})) \equiv \mathcal{L}_{\text{mean}}(\phi, \{h_j\}_{j=1}^K, \{g_j\}_{j=1}^K; \mathcal{D}) \tag{4}$$

$$+ \mathcal{L}_{\text{match}}(\phi, \{h_j\}_{j=1}^K, \{g_j\}_{j=1}^K; \mathcal{D}) \tag{5}$$

$$+ \lambda \, \mathcal{L}_{\text{match}}(\phi, \{h_j\}_{j=1}^K, \{g_j\}_{j=1}^K; \mathbf{Ex}(\mathcal{E})), \tag{6}$$

where $\mathcal{E}$ is our set of points to expand (e.g., using Alg. 2). Note that we provide additional supervisory losses on non-augmented $\mathcal{D}$ via $\mathcal{L}_{\text{mean}}$. In practice, we considered simple linear heads. At an intuitive level, this forces the MLP to learn a robust feature embedding that can 'mimic' the diverse views that the base experts provide. Empirical results show (see Sec. 4) that the network heads learn an effective (often better) estimator than the base expects. However, we see more consistent improvements by not forgetting the base experts and bagging as:

$$f_{\text{EMOE}}(x) \equiv \frac{1}{2K} \sum_{j=1}^K g_j(x) + h_j(\phi(x)). \tag{7}$$

**Motivation** Below we include high-level hypotheses on how the EMOE approach may learn better estimates on OOD data through *variance reduction and regularization*. Previous work has decomposed OOD generalization into bias/variance terms (Yang et al., 2020; Arpit et al., 2022b):

$$\mathbb{E}_{(x,y)\sim\mathcal{P}_{\text{out}}}\mathbb{E}_{\mathcal{D}\sim\mathcal{P}_{\text{in}}}[\text{CE}(y, f(x;\mathcal{D}))] = \mathbb{E}_{(x,y)}[\text{CE}(y, \bar{f}(x))] + \mathbb{E}_{x,\mathcal{D}}[\text{KL}(\bar{f}(x), f(x;\mathcal{D}))] \quad (8)$$

where CE is the cross-entropy loss, $f(x;\mathcal{T})$ is the model fit on dataset $\mathcal{T}$, $\bar{f}(x) = \mathbb{E}_{\mathcal{D}}[f(x;\mathcal{D})]$ is the expected prediction when averaging out draws on the (in-distribution) training dataset $\mathcal{D}$, and $\mathcal{P}_{\text{out}}$ is the OOD data distribution at inference time. Letting $\bar{g}(x) \equiv \frac{1}{K}\sum_{j=1}^{K} g_j(x)$, we may view $\bar{g}(x)$ as a bootstrap-like estimate for $\bar{f}(x)$. One may then take $\mathbb{E}_{x,\mathcal{D}}[\text{KL}(\bar{g}(x), f(x;\mathcal{D}))]$ as a proxy for $\mathbb{E}_{x,\mathcal{D}}[\text{KL}(\bar{f}(x), f(x;\mathcal{D}))]$ and roughly consider

$$\mathbb{E}_{(x,y)\sim\mathcal{P}_{\text{out}}}\mathbb{E}_{\mathcal{D}\sim\mathcal{P}_{\text{in}}}[\text{CE}(y, f(x;\mathcal{D}))] \approx \mathbb{E}_{(x,y)}[\text{CE}(y, \bar{f}(x))] + \mathbb{E}_{x,\mathcal{D}}[\text{KL}(\bar{g}(x), f(x;\mathcal{D}))], \quad (9)$$

which connects to (eq. 5) when interpreting our expanded points as a proxy for the OOD distribution $\mathcal{P}_{\text{out}}$ and $\mathcal{L}_{\text{match}}(\phi, \{h_j\}_{j=1}^K, \{g_j\}_{j=1}^K; \mathbf{Ex}(\mathcal{E}))$ as a proxy for $\mathbb{E}_{x,\mathcal{D}}[\text{KL}(\bar{g}(x), f(x;\mathcal{D}))]$.

## 4 EXPERIMENTS

We conduct experiments on a varied set of real-world datasets to test the OOD generalizability of EMOE. We considered the single source domain generalization setting (e.g., (Qiao et al., 2020)), where our model is trained solely on ID data without any (labeled or unlabelled) OOD data during training/validation (e.g., precluding typical semi-supervised approaches), and without any accompanying environmental/domain/source information from ID training instances. Moreover, we note that we avoided utilizing any modality-specific information in EMOE (e.g., we do not utilize any domain specific augmentations) for generality. We utilized XGB Classifiers (Chen and Guestrin, 2016) fitted to random subsets of data instances and features as the base collection of experts. For a fair/realistic evaluation, we avoided any hyper-parameter tuning on EMOE and utilized a fixed architecture of a 2 layer 512 ELU (Clevert et al., 2015) hidden-unit MLP with 1024 linear-output heads (please see other hyperparameters in Supp. Mat.A.1). For our latent space, we utilize PCA with 128 components. While OOD generalization is an active field of research (Freiesleben and Grote, 2023; Liu et al., 2021), methodology for general (non-modality specific) single source domain generalization is more limited. Given that EMOE integrates numerous techniques, we provide context for our results and compared EMOE with existing strong domain generalization methods that approach the problem from various perspectives and strategies (and are applicable in the single-source setting).

In our experiments, we include two baselines that utilize data augmentation: AdvStyle (Zhong et al., 2022) and Mixup (Zhang et al., 2018), as well as two baselines employing ensemble methods: D-BAT (Pagliardini et al., 2023) and EoA (Arpit et al., 2022a). Mixup linearly combines two ID samples, AdvStyle adversarially augments ID data, D-BAT enforces prediction diversity on OOD data, and EoA ensembles moving average models. For prediction thresholding (rejection), we directly utilize the conditional probability $P(Y = 1 \mid X = x)$ generated by the models. Many real-world applications (e.g. in drug discovery and virtual screen, see Fig. 2) shall utilize only high-confidence predictions. Thus, alongside AUPRC and AUROC, we paid close attention to high-confidence filtration and reported percent of AUPRC at conservative recall thresholds (e.g., 'AURPC@R≤.2'). This metric is computed as the area under the PR-curve up to the specified recall threshold and dividing by the maximum possible area for that threshold (i.e., the threshold). We report both base expert ('EMOE Base') and EMOE (eq. 7) ensemble performance. Our implementation shall be open-sourced upon publication.

### 4.1 CHEMICAL DATASETS

To test how well EMOE generalizes to data with domain shifts in chemical domains, we considered a total of seven datasets from ChEMBL (Gaulton et al., 2011), Therapeutics Data Commons (Huang et al., 2021), and DrugOOD (Ji et al., 2022). For all datasets, we represented molecules using extended-connectivity fingerprints (Rogers and Hahn, 2010) with radius 2 (ECFP4) and with dimensionality 1024. ECFP4 is a standard method for molecular representation and was chosen for its simplicity in calculation as well as its ability to perform comparably to learned representations, such as those generated by graph neural networks on relevant classification tasks (Zagidullin et al.,

2021). Datasets for inhibition of human Ether-à-go-go-Related Gene (hERG), cytotoxicity of human A549 cells (A549_cells), and agonists for Cytochrome P450 2D6 (cyp_2D6) were collected from ChEMBL (Gaulton et al., 2011). For these datasets, binary classification labels were generated using a pChEMBL threshold of 5.0. We also considered an additional binary classification dataset for Ames mutagenicity (Ames) that was taken from Therapeutics Data Commons (TDC) (Huang et al., 2021). For the ChEMBL and TDC datasets (hERG, A549_cells, cyp_2D6, and Ames), ID and OOD splits were determined based on the Murko scaffold of a molecule, such that OOD data have molecular scaffolds not present in the ID data, mimicking the "scaffold domain" approach utilized in DrugOOD (Ji et al., 2022).

Moreover, we considered the "core ec50," "refined ec50," and "core ic50" ligand-based affinity prediction datasets (lbap) from DrugOOD (Ji et al., 2022) (the three hardest OOD performance gap datasets). For these datasets ID and OOD splits were determined based on size, or the number of atoms in a molecule, such that larger molecules are considered the OOD set and smaller molecules, the ID set. Datasets are organized by domain and subsequently divided into training, OOD validation, and OOD testing sets in sequential order. Hence, the OOD validation set from DrugOOD differs in distribution from the OOD testing set. While larger in samples, the DrugOOD datasets ignore the impact of biological targets on label, resulting in a modeling task that has limited relevance to drug discovery. The additional ChEMBL and TDC datasets, though smaller, have direct relevance for drug

Table 1: Experiment results on ChEMBL (Gaulton et al., 2011) and Therapeutics Data Commons (Huang et al., 2021) datasets. We **bold** best scores based on the mean minus 1 standard deviation.

| | | hERG | A549_cells | cyp_2D6 | Ames |
|---|---|---|---|---|---|
| AUPRC@ R<0.2 | D-BAT | 84.48±3.90 | 98.26±0.32 | 91.40±2.20 | 99.04±0.53 |
| | AdvStyle | 88.21±1.73 | 97.77±0.63 | 84.83±2.08 | **99.05±0.38** |
| | EoA | 63.80±0.94 | 61.31±0.70 | 61.77±0.92 | 78.74±0.96 |
| | Mixup | 82.25±3.37 | 95.04±0.56 | 87.09±5.18 | 91.02±2.36 |
| | EMOE base | 94.49±0.54 | 98.29±0.24 | 94.96±0.41 | 98.14±0.22 |
| | EMOE | **95.18±0.79** | **98.95±0.22** | **96.38±0.65** | 98.66±0.37 |
| AUPRC | D-BAT | 54.60±3.55 | 67.04±1.18 | 47.42±1.91 | 70.44±1.70 |
| | AdvStyle | 51.54±2.08 | 65.02±1.60 | 44.41±2.15 | 74.98±1.09 |
| | EoA | 43.30±1.15 | 44.95±0.54 | 37.37±2.12 | 59.43±0.41 |
| | Mixup | 42.42±1.91 | 50.52±1.11 | 27.79±3.34 | 60.94±1.78 |
| | EMOE base | 72.51±0.23 | 84.09±0.10 | 72.72±0.20 | 87.50±0.07 |
| | EMOE | **73.73±0.42** | **84.67±0.09** | **73.77±0.32** | **88.53±0.19** |
| AUROC | D-BAT | 76.58±1.01 | 78.16±0.51 | 67.54±1.06 | 83.82±0.34 |
| | AdvStyle | 75.84±1.02 | 76.13±0.62 | 65.51±1.39 | **85.56±1.59** |
| | EoA | 68.02±0.76 | 68.33±0.53 | 60.50±1.07 | 74.77±0.55 |
| | Mixup | 73.96±0.57 | 76.57±0.94 | 67.53±2.02 | 78.43±1.09 |
| | EMOE base | 74.87±0.10 | 79.17±0.07 | 70.30±0.20 | 81.86±0.11 |
| | EMOE | **76.16±0.28** | **79.54±0.07** | **70.54±0.42** | 83.59±0.24 |

Table 2: Experiment results on DrugOOD (Ji et al., 2022) datasets. We **bold** best scores based on the mean minus 1 standard deviation.

| | | core ec50 val | core ec50 test | refined ec50 val | refined ec50 test | core ic50 test | core ic50 test |
|---|---|---|---|---|---|---|---|
| AUPRC@ R<0.2 | D-BAT | 93.81±0.49 | **84.35±3.01** | 96.97±0.36 | 88.78±0.90 | 98.13±0.19 | 91.79±0.84 |
| | AdvStyle | 94.84±0.69 | 84.51±5.27 | 95.13±0.29 | 88.21±0.83 | 97.04±0.38 | 89.05±0.50 |
| | EoA | 81.85±0.53 | 71.84±1.01 | 85.03±0.14 | 78.79±0.32 | 88.56±0.12 | 77.03±0.29 |
| | Mixup | 83.97±1.37 | 73.04±0.94 | 85.39±0.52 | 79.78±0.75 | 88.99±0.96 | 78.07±1.36 |
| | EMOE base | 97.88±0.30 | 68.91±0.57 | 98.18±0.22 | 89.38±0.68 | **99.13±0.02** | 94.10±0.26 |
| | EMOE | **98.56±0.19** | 70.68±1.04 | **98.22±0.15** | 89.99±0.63 | 99.11±0.07 | **94.45±0.17** |
| AUPRC | D-BAT | 76.64±1.10 | 54.87±2.21 | 84.70±1.15 | 70.08±1.55 | 90.84±0.62 | 73.45±2.02 |
| | AdvStyle | 81.17±3.85 | 58.40±4.88 | 83.01±2.01 | 69.48±4.83 | 88.54±3.22 | 72.11±3.33 |
| | EoA | 64.16±1.14 | 36.50±3.30 | 69.66±1.06 | 57.71±1.77 | 79.12±0.21 | 56.52±0.95 |
| | Mixup | 73.03±3.73 | 60.84±9.64 | 80.36±1.96 | 72.88±3.73 | 86.88±0.32 | 74.99±0.33 |
| | EMOE base | 88.48±0.10 | **71.94±0.13** | 91.26±0.08 | 82.55±0.18 | 94.87±0.04 | 84.14±0.11 |
| | EMOE | **89.58±0.13** | 71.55±0.50 | **91.59±0.07** | **83.27±0.32** | **95.31±0.04** | **84.77±0.08** |
| AUROC | D-BAT | 75.26±0.62 | 58.21±0.58 | **72.09±0.43** | 60.32±0.56 | **80.31±0.18** | 64.82±0.41 |
| | AdvStyle | 75.97±0.88 | **58.86±0.55** | 70.78±0.78 | 59.62±0.68 | 78.36±0.52 | 64.14±0.60 |
| | EoA | 64.91±0.75 | 52.71±0.98 | 59.27±0.44 | 54.63±0.54 | 62.99±0.35 | 55.83±0.41 |
| | Mixup | 68.20±1.48 | 56.33±1.00 | 60.39±0.90 | 56.50±0.82 | 64.24±2.75 | 57.75±1.79 |
| | EMOE base | 73.69±0.09 | 56.58±0.09 | 70.26±0.10 | 59.72±0.19 | 77.66±0.11 | 64.93±0.12 |
| | EMOE | **75.69±0.20** | 54.62±0.69 | 71.47±0.22 | **60.90±0.63** | 79.77±0.13 | **66.04±0.15** |

Table 3: Experiment results on PACS (Li et al., 2017) and Tableshift (Gardner et al., 2023) datasets. We **bold** best scores based on the mean minus 1 standard deviation.

| | | PACS dog-elephant | PACS giraffe-horse | Childhood Lead | FICO HELOC | Hospital Readmission | Sepsis |
|---|---|---|---|---|---|---|---|
| AUPRC@ @R≤.2 | D-BAT | 58.35±7.27 | 80.20±2.58 | 62.82±0.00 | 91.20±0.54 | **78.84±0.28** | 75.37±0.85 |
| | AdvStyle | 55.83±5.64 | 84.77±6.41 | 64.96±0.02 | 88.71±1.46 | 72.91±1.29 | 59.83±1.41 |
| | EoA | 45.16±6.39 | 68.53±4.25 | 77.43±2.17 | 59.53 ±5.02 | 51.83±3.71 | 41.10±3.50 |
| | Mixup | 56.46±6.74 | 81.51±5.45 | 50.00±0.00 | 91.16±4.70 | 69.19±8.84 | 66.09±2.57 |
| | EMOE base | 60.36±0.39 | 82.86±0.36 | 97.21±0.53 | 89.84±0.62 | 58.30±0.21 | **84.27±3.95** |
| | EMOE | **61.94±1.82** | **84.10±1.44** | **97.77±0.82** | **92.12±1.17** | 67.57±0.26 | 82.35±3.24 |
| AUPRC | D-BAT | 54.27±2.78 | 66.10±1.74 | 71.85±0.02 | 80.91±0.39 | 63.29±0.19 | 58.17±0.48 |
| | AdvStyle | 53.52±2.00 | 67.94±3.99 | 48.37±6.19 | 79.63±3.70 | 38.13±8.49 | 54.34±0.60 |
| | EoA | 45.42±5.95 | 68.40±4.41 | 49.48±0.42 | 59.53±5.02 | 29.45±11.08 | 11.21±4.94 |
| | Mixup | 54.05±1.80 | 67.99±2.66 | 50.00±0.00 | 80.95±1.40 | 14.15±2.37 | 56.80±0.89 |
| | EMOE base | 54.47±0.01 | **73.10±0.64** | 85.79±0.13 | 83.73±0.13 | 62.83±0.07 | **67.16±2.75** |
| | EMOE | **56.64±1.47** | 72.76±1.25 | **85.85±0.37** | 83.99±0.28 | 63.62±0.19 | 63.53±1.96 |
| AUROC | D-BAT | 56.44±2.66 | 63.96±2.38 | 79.13±0.04 | 76.13±0.07 | 63.22±0.07 | 57.95±0.08 |
| | AdvStyle | 56.24±1.28 | 65.88±3.53 | 74.45±0.08 | 77.23±3.78 | 61.32±0.77 | 55.31±0.75 |
| | EoA | 49.82±0.92 | 54.93±3.95 | 72.62±0.72 | 54.67±5.71 | 51.65±3.81 | 49.14±5.82 |
| | Mixup | 57.39±1.16 | 64.12±2.78 | 50.00±0.00 | 78.74±0.38 | 63.37±0.55 | 56.82±0.58 |
| | EMOE base | 56.94±0.15 | **72.86±0.61** | 84.45±0.41 | **83.50±0.08** | 63.18±0.06 | **65.13±2.65** |
| | EMOE | **59.99±2.12** | 72.38±1.18 | **87.16±0.23** | 82.98±0.11 | **63.65±0.16** | 61.53±1.58 |

discovery tasks. In our experiments, we assessed performance on both of these sets. Our results are shown in Table 1 and Table 2.

## 4.2 Other Real World Datasets

Next, we further evaluate our method in non-chemical domains, and tested across a diverse range of real-world OOD scenerios using both the Tableshift datasets (Gardner et al., 2023) and images from the Photo-Art-Cartoon-Sketch (PACS) dataset (Li et al., 2017). We selected a diverse collection of Tableshift datasets, based on unrestricted availability and in/out-of-domain performance discrepancy, coverings areas including: finance, education, and healthcare. Each dataset has an associated real-world shift and a related prediction target (see (Gardner et al., 2023) for further details). The PACS dataset includes images from four distinct domains: photo, art painting, cartoon, and sketch. Specifically, we focus on the animal classes (dog, elephant, giraffe, and horse) to create 2 challenging binary classification tasks. Models were trained on the 'photo', 'art', and 'cartoon' domains; they were tested on the unseen fourth domain, 'sketch', to assess generalization performance. Results on the PACS and Tableshift are shown in Tab. 3. As before, we consider the same single-source domain generalization setting. We can see that even over diverse applications, our EMOE method is able to perform well and is often outperforming our strong competing baselines.

## 4.3 Ablation Studies

We empirically validate our per expert matching, and trail-based filtration (Sec. 3.2) with ablations.

**Matching Ablations** We begin by ablating the matching scheme on base experts and explore a mean-only matching approach on expanded points as an alternative. First, we consider a similar training scheme to EMOE (eq. 5), but utilizing a single-headed (SH) MLP, $f(x)$ $(512 \rightarrow 512 \rightarrow 1)$, which is trained via a mean matching loss $\mathcal{L}_{\mathrm{MM}}(f, \{g_j\}_{j=1}^K; \mathcal{S}) \equiv \frac{1}{|\mathcal{S}|} \sum_{x \in S} \ell(f(x), \frac{1}{K} \sum_{j=1}^K g_j(x))$, rather than the per-expert matching loss, $\mathcal{L}_{\mathrm{match}}$ (eq. 2) We also explored the effect of training our multi-headed (MH) architecture $(512 \rightarrow 512 \rightarrow 1024)$ using mean-matching, $\mathcal{L}'_{\mathrm{MM}}(\{h_j\}_{j=1}^K, \{g_j\}_{j=1}^K; \mathcal{S}) \equiv \frac{1}{|\mathcal{S}|} \sum_{x \in S} \ell(\frac{1}{K} \sum_{j=1}^K \sigma(h_j(x)), \frac{1}{K} \sum_{j=1}^K g_j(x))$. Lastly, we compare these ablations to our EMOE approach which trains the multi-headed architecture with per-expert matching (eq. 5). We report results as the average change ($\Delta$) in performance for AUPRC using the EMOE predictions (eq. 7) vs the base experts at various recall thresholds. (Greater values indicate

Table 4: ChEMBL datasets' mean $\Delta$AUPRC over base experts ablating augmentation.

| | Δ@R<.2 | Δ@R<1 |
|---|---|---|
| SH MLP + MM | 0.59 | 0.45 |
| MH MLP + MM | 0.25 | 0.55 |
| EMOE | 1.11 | 1.03 |

greater improvement over the base experts.) As seen in Tab. 4, individual expert matching yielded the best results.

**Trial Based Filtration**   Next we ablate our augmentation strategy. We considered an alternative generic confidence based strategy ('Conf.'), that expands the dataset on randomly drawn points and also performs random convex combinations of pairs of points. These points are then labeled with the EMOE base experts and those with high confidence are kept for matching. In contrast, in our EMOE approach we only expand those points that were contained in successful trials (see Sec. 3.2), and filter them further based on confidence. Although our approach is stable w.r.t. different augmentation strategies we see the best results by incorporating our extrapolatory trials for filtration as shown in Tab. 5.

Table 5: ChEMBL datasets' mean $\Delta$AUPRC over base experts ablating augmentation.

|  | $\Delta$@R<.2 | $\Delta$@R<1 |
|---|---|---|
| EMOE Conf. | 0.93 | 0.96 |
| EMOE | 1.11 | 1.03 |

### 4.4   DISCUSSION

Below we expound on major takeaways from our results. First, it is worth noting the base-experts are providing relatively competitive performance compared to the more complicated baselines. This motivates our philosophy of building on, and strengthening, the predictions of the base experts. Moreover, we see consistent improvements by EMOE, indicated by its leading tally of 33 across all metrics/datasets compared to the next highest score of 4 from the strong competing baseline D-Bat (Pagliardini et al., 2023) as shown in Tab. 6. Despite its elevated performance, EMOE does not incur an out-sized computational cost. For example on the 'hERG' dataset, the wall clock time for an unoptimized implementation of EMOE's base models, and neural network amounts to 10.8 (which can parallelized for further efficiency) and 23.2 minutes respectively, which places it between the quicker baselines like Mixup (1.2 minutes) and D-Bat (3.5 minutes) and the slower ones like EoA (67.9 minutes).

EMOE especially stands out for its precision at lower recall values; we can visualize reasons for the improvement in performance as follows. As can be seen in the scatter plot

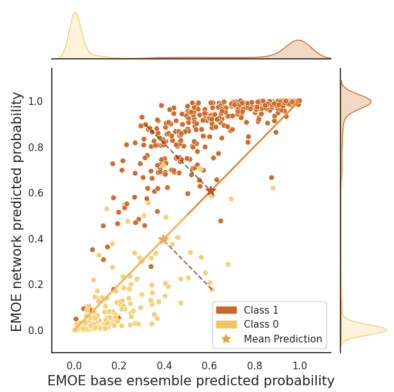

Figure 6: *Mean predicted probabilities from EMOE and EMOE base model.* Starred points denote instances where the base model initially makes incorrect predictions but are corrected when we average the predicted probabilities from EMOE and EMOE base.

(Fig. 6) the multi-headed network rectifies several of the base expert's mistakes on OOD data.

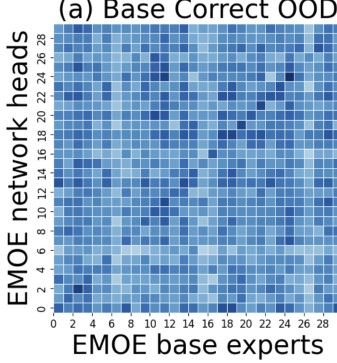 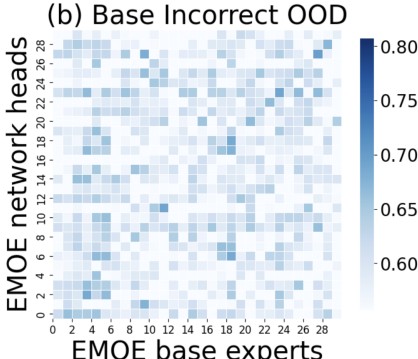

Figure 5: *Experts correlations between EMOE base experts and EMOE network heads on "Ames" dataset.* (a) EMOE experts has a high correlation with EMOE base experts on sample the base experts makes correct predictions. (b) EMOE experts shown low correlations with EMOE base experts on sample the base experts makes incorrect predictions.

Moreover, it can be seen that the multi-headed network is also increasing the certainty of predictions on OOD data as predictions are moving away from $0.5$. It is well-known that diversity in ensembles improve performance (Fort et al., 2019); however, we want diversity of ensemble on potentially erroneous predictions. This is exactly the behavior that we observe in Fig. 5; while the heads of the EMOE network show significant agreement on OOD samples where the base experts make correct predictions, they display significantly less agreement (more diversity) on OOD samples where base experts make incorrect predictions. This diversity contributes to improved performance.

Table 6: Experiment results summary. Number of datasets where respective methods lead in metrics (tallied from tables above).

|  |  | EMOE | EMOE base | D-Bat | Advstyle | EoA | Mixup |
|---|---|---|---|---|---|---|---|
| metric | AUPRC@R<0.2 | **11** | 2 | 2 | 1 | 0 | 0 |
|  | AUPRC | **13** | 3 | 0 | 0 | 0 | 0 |
|  | AUROC | **9** | 3 | 2 | 2 | 0 | 0 |
|  | Total | **33** | 8 | 4 | 3 | 0 | 0 |

## 5 CONCLUSION

In summary, this work presents Expansive Matching on Experts (EMOE), a novel method that utilizes support-expanding, extrapolatory pseudo-labeling to improve prediction and uncertainty based rejection on OOD points. Our techniques are general and not specific to any data-modality, nor do they require additional unlabeled data or domain information. Moreover, they are largely complementary to other existing approaches. Thus, we envision the potential for future work that incorporates our methodology jointly with other OOD generalization techniques. Moreover, we are encouraged by results in chemical-property prediction and shall further explore future work in these directions.

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

# A    APPENDIX

## A    ADDTIONAL EXPERIMENT DETAILS

### A.1    EMOE TRAINING DETAILS

In all of our experiments we used the Adam (Kingma and Ba, 2014) optimizer and mini-batches of size 256. One Nvidia A100 GPU with 40GB GPU memory was used to run our experiments, and duration for model training is approximately 0.5 hours. During the extrapolatory directional mining trials, we held out 0.1 of the data for each of the 1000 trials. Subsequently, from the top-performing trials, determined by accuracy evaluation, we selected the top held-out points from 0.15 of the best-performing trials for data expansion. $\lambda = 0.5$ was used for the $\mathcal{L}_{\text{match}}$ for the expanded points. As noted in Sec. 3.1 we trained the EMOE models directly in the latent space to avoid the need for the decoder (and also allowed baselines to do this if it aided their performance). In the experiments on hERF, A549_cells, CYP_2D6, Ames, core ec50, refined ec50, EMOE was trained for 20000 iterations. Arithmetic mean between EMOE and EMOE base was reported. On the M/C-D and M/F-D datasets, we took the raw pixel values as input and trained EMOE networks for 4000 iterations due to lower observed losses. Harmonic mean between EMOE and EMOE base was reported on these datasets. We performed 8 trails on each of the datasets for EMOE.

### A.2    BASELINE SETUP

We re-implemented all baselines we are comparing against EMOE following the implementation details in their paper and/or using Github implementations (if available). Since the fingerprints representation of chemicals are quite sparse, we preformed dimension reduction using PCA with 128 components on all chemical datasets. For D-BAT(Pagliardini et al., 2023) with existing implementations designed for tabular data, we utilized their original model architectures. For the other three baseline methods without implementation specifically for tabular data, we adopted a structure comprising two 512 ELU(Clevert et al., 2015) layers to closely mimic the EMOE network architecture. The Adam (Kingma and Ba, 2014) optimizer was used for training baseline models.

**D-BAT** In our experiments, the D-Bat(Pagliardini et al., 2023) models used MLP architecture with one 128 LeakyRelu(Maas, 2013) layer following the architecture in their Github. Their paper (Pagliardini et al., 2023) discussed two settings, and we focused on the scenario where perturbation data differs from the distribution of test data, adhering to the single-source domain generalization setting. We trained an ensemble of five models sequentially for the D-bat baseline models and the predictions from the 5 models were averaged to obtain the final prediction.

**EoA** We trained an ensemble of 5 simple moving average model following the method described in (Arpit et al., 2022a). We start calculating the moving average at iteration 50 and trained the models for 200 iterations. The predictions from the 5 models were averaged to obtain the final prediction for EoA.

For AdvStyle (Zhong et al., 2022) and Mixup(Zhang et al., 2018), the methodologies were straight-forward. We experimented with training using various numbers of iterations and reported the most promising results. Note that we used alpha=0.7 when combining the 2 samples for Mixup. We executed all baseline experiments five times on each dataset to ensure a precise estimation of performance.

### A.3    EXAMPLES OF PACS DATASETS

## B    ADDITIONAL EXPERIMENT AND ABLATION RESULTS

### B.1    FULL EXPERIMENT RESULTS ON CHEMBL AND THERAPEUTICS DATA COMMONS

In Table 7, we report the full results on hERG, A549_cells, cyp_2D6. and Ames.

Table 7: Full experiment results on ChEMBL (Gaulton et al., 2011) and Therapeutics Data Commons (Huang et al., 2021) datasets. We **bold** best scores based on the mean minus 1 standard deviation.

|  |  | hERG | A549_cells | cyp_2D6 | Ames |
|---|---|---|---|---|---|
| AUPRC @R≤.1 | D-BAT | 88.55±3.75 | 98.57±0.36 | 95.71±1.99 | 99.07±0.52 |
|  | AdvStyle | 93.27±1.25 | 96.89±0.66 | 84.21±4.62 | 99.52±0.26 |
|  | EoA | 63.80±0.94 | 61.31±0.70 | 61.77±0.92 | 78.74±0.96 |
|  | Mixup | 82.80±3.49 | 95.04±0.56 | 87.39±6.91 | 91.02±2.36 |
|  | EMOE base | 96.69±0.62 | 99.79±0.11 | **99.55±0.58** | 99.30±0.25 |
|  | EMOE | **98.98±0.56** | **99.85±0.14** | 99.26±0.59 | **99.76±0.37** |
| AUPRC@ @R≤.2 | D-BAT | 84.48±3.90 | 98.26±0.32 | 91.40±2.20 | 99.04±0.53 |
|  | AdvStyle | 88.21±1.73 | 97.77±0.63 | 84.83±2.08 | **99.05±0.38** |
|  | EoA | 63.80±0.94 | 61.31±0.70 | 61.77±0.92 | 78.74±0.96 |
|  | Mixup | 82.25±3.37 | 95.04±0.56 | 87.09±5.18 | 91.02±2.36 |
|  | EMOE base | 94.49±0.54 | 98.29±0.24 | 94.96±0.41 | 98.14±0.22 |
|  | EMOE | **95.18±0.79** | **98.95±0.22** | **96.38±0.65** | 98.66±0.37 |
| AUPRC@ @R≤.3 | D-BAT | 82.44±3.56 | 97.37±0.53 | 87.65±1.29 | 98.61±0.53 |
|  | AdvStyle | 85.05±1.93 | 96.47±0.64 | 82.76±2.13 | **98.71±0.44** |
|  | EoA | 63.80±0.94 | 61.31±0.70 | 61.77±0.92 | 78.74±0.96 |
|  | Mixup | 81.51±3.01 | 94.95±0.55 | 84.53±6.46 | 90.88±2.24 |
|  | EMOE base | 91.05±0.50 | 97.34±0.21 | 91.60±0.32 | 98.05±0.15 |
|  | EMOE | **91.57±0.61** | **98.11±0.25** | **93.10±0.81** | 98.41±0.29 |
| AUPRC | D-BAT | 54.60±3.55 | 67.04±1.18 | 47.42±1.91 | 70.44±1.70 |
|  | AdvStyle | 51.54±2.08 | 65.02±1.60 | 44.41±2.15 | 74.98±1.09 |
|  | EoA | 43.30±1.15 | 44.95±0.54 | 37.37±2.12 | 59.43±0.41 |
|  | Mixup | 42.42±1.91 | 50.52±1.11 | 27.79±3.34 | 60.94±1.78 |
|  | EMOE base | 72.51±0.23 | 84.09±0.10 | 72.72±0.20 | 87.50±0.07 |
|  | EMOE | **73.73±0.42** | **84.67±0.09** | **73.77±0.32** | **88.53±0.19** |
| AUROC | D-BAT | 76.58±1.01 | 78.16±0.51 | 67.54±1.06 | 83.82±0.34 |
|  | AdvStyle | 75.84±1.02 | 76.13±0.62 | 65.51±1.39 | **85.56±1.59** |
|  | EoA | 68.02±0.76 | 68.33±0.53 | 60.50±1.07 | 74.77±0.55 |
|  | Mixup | 73.96±0.57 | 76.57±0.94 | 67.53±2.02 | 78.43±1.09 |
|  | EMOE base | 74.87±0.10 | 79.17±0.07 | 70.30±0.20 | 81.86±0.11 |
|  | EMOE | **76.16±0.28** | **79.54±0.07** | **70.54±0.42** | 83.59±0.24 |

## B.2 FULL EXPERIMENT RESULTS ON DRUGOOD

In Table 8, we report the full results on core ec50, refined ec 50, and core ic50 from DrugOOD (Ji et al., 2022).

## B.3 FULL EXPERIMENT RESULTS ON PACS DATASET

In Table 9, we report the full results on images from PACS (Li et al., 2017).

## B.4 ABLATIONS RESULTS ON VANILLA AND MOE MODELS WITHOUT ENSEMBLE BASE MODEL

In this ablation study, we focused on examining the impact of the vanilla vs. Mixture of Experts (MoE) architecture on OOD performance. Both the MoE MLP and Vanilla MLP were configured with two layers of 512 ELU units. The MoE MLP has 1024 output heads, whereas the vanilla MLP had only one. For noisy augmentation, we applied small perturbations drawn from a standard normal distribution to the training set while retaining the original labels. We run 3 trials on each of the CheMBL dataset and the results are shown in Table 10 . Although we observed that the MoE model achieved a AUPRC compared to the vanilla MLP, we found that simple noisy augmentation did not lead to any significant difference in performance.

## C LIMITATIONS

It is important to acknowledge a reliance on the EMOE base model and the latent space; we observed good performance with simple implementations, indicating the potential for better performance with other choices. Moreover, to preserve generality, this work limited itself to augmentations in a real-value vector setting; however, when it may be possible to exploit modality-specific augmentations in applications.

Table 8: Full experiment results on DrugOOD datasets. We **bold** best scores based on the mean minus 1 standard deviation.

| | | core ec50 val | core ec50 test | refined ec50 val | refined ec50 test | core ic50 test | core ic 50 test |
|---|---|---|---|---|---|---|---|
| AUPRC@ @R≤.1 | D-BAT | 94.04±0.55 | **86.59±3.17** | 97.19±0.43 | 88.93±1.07 | 98.25±0.21 | 91.89±0.86 |
| | AdvStyle | 95.79±0.45 | 84.56±5.28 | 96.37±0.72 | 88.69±0.95 | 98.10±0.37 | 89.39±0.73 |
| | EoA | 81.85±0.53 | 71.84±1.01 | 85.03±0.14 | 78.79±0.32 | 88.56±0.12 | 77.03±0.29 |
| | Mixup | 83.97±1.37 | 73.03±0.93 | 85.39±0.52 | 79.78±0.75 | 88.99±0.96 | 78.07±1.36 |
| | EMOE base | 98.85±0.33 | 66.19±1.14 | **99.09±0.11** | **92.72±0.66** | **99.56±0.01** | **96.85±0.17** |
| | EMOE | **99.05±0.26** | 68.12±1.16 | 98.85±0.20 | 91.26±0.80 | 99.38±0.08 | 96.37±0.31 |
| AUPRC@ @R≤.2 | D-BAT | 93.81±0.49 | **84.35±3.01** | 96.97±0.36 | 88.78±0.90 | 98.13±0.19 | 91.79±0.84 |
| | AdvStyle | 94.84±0.69 | 84.51±5.27 | 95.13±0.29 | 88.21±0.83 | 97.04±0.38 | 89.05±0.50 |
| | EoA | 81.85±0.53 | 71.84±1.01 | 85.03±0.14 | 78.79±0.32 | 88.56±0.12 | 77.03±0.29 |
| | Mixup | 83.97±1.37 | 73.04±0.94 | 85.39±0.52 | 79.78±0.75 | 88.99±0.96 | 78.07±1.36 |
| | EMOE base | 97.88±0.30 | 68.91±0.57 | 98.18±0.22 | 89.38±0.68 | **99.13±0.02** | 94.10±0.26 |
| | EMOE | **98.56±0.19** | 70.68±1.04 | **98.22±0.15** | **89.99±0.63** | 99.11±0.07 | **94.45±0.17** |
| AUPRC@ @R≤.3 | D-BAT | 93.73±0.48 | **81.40±2.00** | 96.89±0.33 | 87.77±0.71 | 98.08±0.19 | 90.71±1.17 |
| | AdvStyle | 94.52±0.82 | 83.57±4.65 | 94.71±0.29 | 88.05±0.81 | 96.69±0.48 | 88.93±0.42 |
| | EoA | 81.85±0.53 | 71.84±1.01 | 85.03±0.14 | 78.79±0.32 | 88.56±0.12 | 77.03±0.29 |
| | Mixup | 83.97±1.37 | 73.06±0.96 | 85.39±0.52 | 79.78±0.75 | 88.99±0.96 | 78.07±1.36 |
| | EMOE base | 96.85±0.28 | 70.30±0.45 | 97.30±0.25 | 87.82±0.53 | 98.69±0.03 | 92.32±0.23 |
| | EMOE | **97.86±0.20** | 71.35±0.87 | **97.49±0.09** | **89.01±0.54** | **98.84±0.07** | **93.11±0.13** |
| AUPRC | D-BAT | 76.64±1.10 | 54.87±2.21 | 84.70±1.15 | 70.08±1.55 | 90.84±0.62 | 73.45±2.02 |
| | AdvStyle | 81.17±3.85 | 58.40±4.88 | 83.01±2.01 | 69.48±4.83 | 88.54±3.22 | 72.11±3.33 |
| | EoA | 64.16±1.14 | 36.50±3.30 | 69.66±1.06 | 57.71±1.77 | 79.12±0.21 | 56.52±0.95 |
| | Mixup | 73.03±3.73 | 60.84±9.64 | 80.36±1.96 | 72.88±3.73 | 86.88±0.32 | 74.99±0.33 |
| | EMOE base | 88.48±0.10 | **71.94±0.13** | 91.26±0.08 | 82.55±0.18 | 94.87±0.04 | 84.14±0.11 |
| | EMOE | **89.58±0.13** | 71.55±0.50 | **91.59±0.07** | **83.27±0.32** | **95.31±0.04** | **84.77±0.08** |
| AUROC | D-BAT | 75.26±0.62 | 58.21±0.58 | **72.09±0.43** | 60.32±0.56 | **80.31±0.18** | 64.82±0.41 |
| | AdvStyle | 75.97±0.88 | **58.86±0.55** | 70.78±0.78 | 59.62±0.68 | 78.36±0.52 | 64.14±0.60 |
| | EoA | 64.91±0.75 | 52.71±0.98 | 59.27±0.44 | 54.63±0.54 | 62.99±0.35 | 55.83±0.41 |
| | Mixup | 68.20±1.48 | 56.33±1.00 | 60.39±0.90 | 56.50±0.82 | 64.24±2.75 | 57.75±1.79 |
| | EMOE base | 73.69±0.09 | 56.58±0.09 | 70.26±0.10 | 59.72±0.19 | 77.66±0.11 | 64.93±0.12 |
| | EMOE | **75.69±0.20** | 54.62±0.69 | 71.47±0.22 | **60.90±0.63** | 79.77±0.13 | **66.04±0.15** |

Table 9: Experiment results on PACS dataset. We **bold** best scores based on the mean minus 1 standard deviation.

| | | PACS dog-elephant | PACS giraffe-horse |
|---|---|---|---|
| AUPRC @R≤.1 | D-BAT | 58.33±7.50 | 82.15±3.10 |
| | AdvStyle | 54.52±7.86 | 88.73±7.30 |
| | EoA | 44.84±6.95 | 69.41±3.66 |
| | Mixup | 65.20±7.80 | 84.07±4.53 |
| | EMOE base | **66.58±1.27** | 83.23±0.71 |
| | EMOE | 64.00±1.73 | **85.90±2.92** |
| AUPRC@ @R≤.2 | D-BAT | 58.35±7.27 | 80.20±2.58 |
| | AdvStyle | 55.83±5.64 | 84.77±6.41 |
| | EoA | 45.16±6.39 | 68.53±4.25 |
| | Mixup | 56.46±6.74 | 81.51±5.45 |
| | EMOE base | 60.36±0.39 | 82.86±0.36 |
| | EMOE | **61.94±1.82** | **84.10±1.44** |
| AUPRC@ @R≤.3 | D-BAT | 57.88±5.46 | 78.21±2.54 |
| | AdvStyle | 56.25±4.83 | 81.76±6.03 |
| | EoA | 45.27±6.21 | 67.57±3.79 |
| | Mixup | 56.33±6.51 | 79.40±4.96 |
| | EMOE base | 58.62±0.07 | **82.69±0.43** |
| | EMOE | **60.80±1.44** | 83.00±1.35 |
| AUPRC | D-BAT | 54.27±2.78 | 66.10±1.74 |
| | AdvStyle | 53.52±2.00 | 67.94±3.99 |
| | EoA | 45.42±5.95 | 68.40±4.41 |
| | Mixup | 54.05±1.80 | 67.99±2.66 |
| | EMOE base | 54.47±0.01 | **73.10±0.64** |
| | EMOE | **56.64±1.47** | 72.76±1.25 |
| AUROC | D-BAT | 56.44±2.66 | 63.96±2.38 |
| | AdvStyle | 56.24±1.28 | 65.88±3.53 |
| | EoA | 49.82±0.92 | 54.93±3.95 |
| | Mixup | 57.39±1.16 | 64.12±2.78 |
| | EMOE base | 56.94±0.15 | **72.86±0.61** |
| | EMOE | **59.99±2.12** | 72.38±1.18 |

Table 10: Ablation results on MLP architectures with CheMBL datasets.

| | AUPRC@R≤.1 | AUPRC@R≤.2 | AUPRC@R≤.3 | AUPRC | AUROC |
|---|---|---|---|---|---|
| Vanilla MLP | 68.96 | 68.96 | 68.96 | 49.73 | 68.26 |
| Vanilla MLP+noisy aug | 69.12 | 69.12 | 69.12 | 48.43 | 67.44 |
| MH_MLP | 70.96 | 70.96 | 70.96 | 53.18 | 69.08 |
| MH_MLP+noisy aug | 70.49 | 70.49 | 70.49 | 54.54 | 69.15 |
| EMOE | 99.04 | 96.21 | 92.90 | 73.73 | 70.59 |

