# OpenReview forum: "EMOE: Expansive Matching of Experts for Robust Uncertainty Based Rejection"
_ICLR.cc/2025/Conference — ICLR 2025 Conference Withdrawn Submission_

### Official Review · Reviewer_hcoy · 2024-11-03

**Soundness:** 2
**Presentation:** 3
**Contribution:** 1
**Rating:** 3
**Confidence:** 4

**Summary:**

The paper presents an approach to identify out-of-distribution (OOD) samples for tabular and sample data. It is proposed to form OOD samples on the feature space with augmentations and then pseudo-label them using a model ensemble. The proposed ensemble is a shared MLP backbone with several heads. In addition, there is a heuristic approach to filter out pseudo-labels that may not be useful.  In this case, a simple parametric model is used to evaluate the pseudo-labels. The approach is extensively evaluated on chemical and tableshifts datasets, as well as on image-based data from the PACS dataset.

**Strengths:**

- The proposed idea is simple and easy to understand. In general, the proposed approach is clearly described.
- There is an extensive evaluation on chemical datasets. The method has been built for this type of data, so it shows that it works well in practice.

**Weaknesses:**

- The major issue of the paper is its limited novelty. It combines existing approaches in a straightforward manner. It does not show new elements even on the existing approaches. This point makes the work weak. The paper would stand easier to an applied ML venue.
- The whole approach and idea is better suited to domain generalisation. It would certainly make sense to present developments on domain generalisation.  The domain evaluation would be more appropriate.
- The PACS dataset is a domain generalisation dataset. It's not clear that it's being used in a different way. It would make more sense to choose a dataset for OOD detection if this is the goal of the paper.
- Another major limitation is the luck of comparison with recent methods. It does not help the paper to show its potential.
- In Section 3.3, the L1 loss for classification is not well motivated. It should be argued why it makes sense to use it for the classification task, e.g. strong regularisation based on the literature.
- Line 178, the reference to "general data" adds ambiguity to the understanding of the approach. It would make more sense to write the paper on tabular data (excluding the images) and explain what the problems are there.

**Questions:**

- It would be interesting to explain why the paper was not written only for tabular data and focused only on domain generalisation.
- What is the reason to rely on PACS for OOD detection and pick up OOD datasets?

---

### Official Review · Reviewer_GrDa · 2024-11-03

**Soundness:** 3
**Presentation:** 3
**Contribution:** 2
**Rating:** 5
**Confidence:** 3

**Summary:**

This paper contributes to enhancing the ability of machine learning models to make predictions and reject uncertain questions when dealing with out-of-distribution data points. To enable model exploration, the authors propose a pseudo-label-based expansive matching method with multiple experts. Experimental results on multiple single-domain generalization benchmarks (i.e. Tableshift, chemical data and etc.) demonstrate the effectiveness of the proposed method.

**Strengths:**

+ The problem of generalizing to OOD data is critical in many real-world applications, especially in medical and drug discovery domains. The paper's focus on improving extrapolation capabilities is timely and could have substantial practical implications.
+ Experimental results are conducted on multiple scenarios (e.g. tabular, chemical and etc), which demonstrate the generalization of the proposed method.

**Weaknesses:**

+ Dependence on Base Experts: While the paper shows improvements over the base experts, there is a reliance on the quality and diversity of these base experts. The performance of EMOE could be limited if the base experts are not sufficiently robust or diverse. Meanwhile, the requirement of multiple experts also increases the computation complexity of the proposed method.

+ The experiments are conducted on simple MLP-based architecture. It would be better that the authors follow [1] and conduct experiments on the more powerful network architecture. For example, the empirical results in [1] show that most domain generalization methods fail to outperform the vanilla baseline when hyperparameters are carefully tuned based on the ResNet-50 and ResNet-101 architectures. It would be beneficial to adopt the network architecture and experimental settings from [1] for the PACS dataset to better demonstrate the effectiveness of the proposed method.

[1] In Search of Lost Domain Generalization

**Questions:**

+ Could the authors provide the qualitative results of the success and failure case during rejection? For example, the authors could visualize the image that the prediction are inaccurate and successfully or fail to reject. This could help the readers better understand the effectiveness and the limitations of the proposed method.

+ What is the value of 'k,' the number of experts used in the experiments? Is the rejection ability sensitive to the number of experts?

+ The paper primarily focuses on single-domain generalization scenarios. However, it is more common for training data to come from multiple domains. Another question is whether the proposed method can effectively generalize to multi-domain scenarios. Take PACS dataset as an example, the authors can take three domains from photo, art painting cartoon and sketch as the training data and the left one as the test set.

---

### Official Review · Reviewer_raUC · 2024-11-04

**Soundness:** 2
**Presentation:** 2
**Contribution:** 2
**Rating:** 3
**Confidence:** 3

**Summary:**

The authors propose a method for label generation by creating OOD instances in a latent space and performing empirical filtering. Specifically, EMOE is a pseudo-label generator equipped with multiple experts, and it demonstrates superior results over baselines on both CHEMICAL DATASETS and TableShift datasets.

**Strengths:**

1. EMOE introduces a novel support-expanding method in latent space to generate OOD instances, improving pseudo-labeling without additional OOD data.
2. The method is data-modality agnostic, working across diverse data types (chemical, tabular, image) without domain-specific adjustments.

**Weaknesses:**

1. The authors mention extrapolatory capability in the introduction, which aligns with the goal of OOD generalization, but their proposed model does not directly address this issue.
2. The normal distribution used in Equation 1 is overly strong and impractical; to validate this assumption, the authors should compare the OOD data generated by their method with real OOD datasets to assess the gap.
3. The authors should conduct experiments on OOD generalization datasets, such as the full spectrum test of OpenOOD, to evaluate their model’s performance against other mainstream OOD methods.

**Questions:**

See Weaknesses

---

### Official Review · Reviewer_U6oJ · 2024-11-07

**Soundness:** 2
**Presentation:** 1
**Contribution:** 2
**Rating:** 5
**Confidence:** 3

**Summary:**

This paper proposes an expansive data augmentation approach, EMOE, to enhance uncertainty-based rejection on out-of-distribution data. EMOE comprises multi-stage training, i.e. initial expert learning and secondary matching model learning.  The novelty lies in the expansive data augmentation and filtering.

**Strengths:**

Expansive data augmentation and filtering are good points for this paper. Experiments have shown some good results.

**Weaknesses:**

This paper is not well written. The presentation of figures and methods is not clear. Fig.1 does not deliver any useful information about the approach. It should show multi-stage training and emphasize the difference between experts and heads. Fig.2 is also not informative, you can consider visualizing the selection on Alg.2. For eq.2-9, please clarify the meaning of g(x) nearby. Currently, g(x) is only shown in 165-167, which makes readers hard to understand these equations. Also, please include math symbols in Alg.1 as your paper defines.

The experiment part is poor. Even the number of experts and the impact of $L_{mean}$ is not verified in experiments.

Please give a clear definition of your problem at the beginning part of the method section. Besides, please clarify what is the meaning of distribution shift here. For example, distribution shifts can be covariate shifts in domain generalization or semantic shifts in out-of-distribution detection.

**Questions:**

Please refer to the weaknesses.

---

### Note · Authors · 2024-12-02

I have read and agree with the venue's withdrawal policy on behalf of myself and my co-authors.